# Iron alloys of volatile elements in the deep Earth's interior

Yifan Tian[1,2], Peiyu Zhang[1,2], Wei Zhang[1,2], Xiaolei Feng[3], Simon A. T. Redfern ®[3,4] & Hanyu Liu ®[1,2,5] ✉

Investigations into the compositional model of the Earth, particularly the atypical concentrations of volatile elements within the silicate portion of the early Earth, have attracted significant interest due to their pivotal role in elucidating the planet's evolution and dynamics. To understand the behavior of such volatile elements, an established 'volatility trend' has been used to explain the observed depletion of certain volatile elements. However, elements such as Se and Br remain notably over-depleted in the silicate Earth. Here we show the results from first-principles simulations that explore the potential for these elements to integrate into *hcp*-Fe through the formation of substitutional alloys, long presumed to be predominant constituents of the Earth's core. Based on our findings, the thermodynamic stability of these alloys suggests that these volatile elements might indeed be partially sequestered within the Earth's core. We suggest potential reservoirs for volatile elements within the deep Earth, augmenting our understanding of the deep Earth's composition.

Investigations of compositional models of the Earth offer a route to understanding the evolution, dynamics and early history of our planet[1,2]. For example, during Earth's core segregation, an important event early in Earth's history, some elements (siderophile, iron-loving) tended to be assimilated into the metallic core, while others (lithophile, rock-loving) remained largely within the silicate mantle[3–8]. For those elements with weak to moderate siderophile tendencies and high condensation temperatures, core-mantle partitioning is clearly delineated, as they are observed in consistent ratios both in the silicate Earth and the chondritic reference[9]. However, concentrations of other volatile elements are less well-matched, showing depletions in the silicate Earth with respect to the chondritic reference.

In view of the high-temperatures prevalent during core-formation, volatility is an important factor for the retention of elements throughout the accretion and core-formation process[10–13]. A known 'volatile trend' line is well established and has been used to explain the abundances of elements in the silicate Earth, with the general observation that the lower the condensation temperature of an element, the greater depletion in the solid Earth[12]. However, by comparing the abundance of the elements with their approximate position in the trend line, some elements appear excessively depleted in the silicate Earth, such as Se and Br, suggesting depletion could be attributed to a combination of both volatility and dissolution or assimilation into the core[14–17]. However, the underlying mechanism of the incorporation of these apparently lithophile elements into the Earth's core remain unresolved from simple considerations of their physical and chemical properties.

It is essential to understand the reactions between liquid iron and non-siderophile or volatile elements under high pressures, since the entire primordial Earth was expected to be molten during core-mantle differentiation[13]. However, computational simulations of such conditions are extremely challenging, and more attention has, therefore, been paid to the study of reactions between solid iron and elements depleted in the silicate earth, e.g. Xe, which is relevant to the understanding of the inner core crystallization after core-mantle differentiation[18]. Recent studies, interestingly, reported the finding

[1]Key Laboratory of Material Simulation Methods and Software of Ministry of Education, College of Physics, Jilin University, Changchun 130012, China. [2]State Key Laboratory of Superhard Materials, College of Physics, Jilin University, Changchun 130012, China. [3]School of Materials Science and Engineering, Nanyang Technological University, 50 Nanyang Avenue, Singapore 639798, Singapore. [4]Asian School of the Environment, Nanyang Technological University, 50 Nanyang Avenue, Singapore 639798, Singapore. [5]International Center of Future Science, Jilin University, Changchun 130012, China. ✉e-mail: hanyuliu@jlu.edu.cn

of a chemical reaction trend of Fe and *p*-block elements by using ab initio structure prediction method under high pressures, and their results show a negative correlation between binding strength with Fe and the depletion of the elements[19]. It is noted that in most scenarios associated with depletion due to assimilation in the core, the depleted elements are expected to partition into the metallic (iron) core as a low solute concentration of iron[20–24]. Moreover, it is generally supposed that the wave velocities observed in seismic data from the core imply an inner core density lower than that of pure iron[25–27]. This suggests the existence of 'light' elements as the major alloying candidates within it, such as H, C, O, S and Si[22,28–42].

In this work, we perform simulations on the reactions between iron and non-siderophile and volatile elements by forming substitutional Fe alloys, where the latter elements were treated as impurities via substitutional incorporation onto Fe sites in *hcp*-Fe supercells. Remarkably, our results reveal that the abundances of non-siderophile and volatile elements in the silicate Earth are positively correlated to the formation enthalpies of these Fe alloys at high pressures, which suggests the likelihood of incorporating such elements into the core. In these reactions, pressure plays a critical role in dramatically altering the thermodynamic stability of these Fe alloys, where lithophile or chalcophile elements tend to become siderophile on increasing pressure. Also, our calculations of elastic properties of the Fe alloys suggest that doping depleted elements into *hcp*-Fe will decrease shear velocity $V_S$ at the inner core pressure, especially for heavy elements, a possibility that has not been fully considered in previous studies that have attempted to explain the density deficit in Preliminary Reference Earth Model (PREM).

## Results

### Stabilities of Fe alloys

As has been discussed previously[9], the elements that appear depleted in the silicate Earth can be ordered in terms of their 50% condensation temperatures at $10^{-4}$ bar, and in terms of their siderophilicities (Table S1). Since the depleted elements with high siderophilicity or low volatility can be presumed to have been sequestered within the deep Earth during the core segregation, here, we focus on the non-siderophile elements with moderate and high volatility (shown in green in Table S1), which are represented in Fig. 1. We have investigated their formation enthalpies upon alloying into *hcp*-Fe at high pressures. Furthermore, we have also included additional elements which are not volatile, such as P and Ge, as shown in Table S1 and Fig. S1, to investigate their possible alloying propensity with Fe at high pressures. In all, we have selected around 40 different elements relevant to the silicate Earth and calculated their enthalpies of formation upon alloying into *hcp*-Fe at 20, 150 and 300 GPa (Fig. S1). The rationale for

selecting these elements to alloy with Fe, rather than forming compounds, is that their quantities are insufficient to generate Fe-compounds with high stoichiometric ratios in the deep Earth. To assess the simulation size employed in our calculations, we conducted simulations using supercells of 54 atoms, as depicted in Fig. S2. Our results indicate that the trend remains consistent regardless of the doping concentration of the depleting elements. Our simulations suggest that pressure can significantly enhance the stability of these Fe alloys, even if some of these impurity atoms exhibit lithophile or chalcophile tendencies at ambient pressures. As Fe gradually accretes into the center of the Earth (pressure increases), the formation enthalpies of these pressure-promoted reactions become lower, so that more elements are available to alloy with *hcp*-Fe. In view of the high temperatures of the deep Earth, we recognize that entropy may play an important role in these reactions. The geotherm of the Earth suggests temperatures of 2000 K, 4000 K and 6000 K (in $TS_{conf}$) correspond to pressures of 20 GPa, 150 GPa and 300 GPa, respectively, (Fig. S3)[43]. Elements locate in shaded areas in Fig. S3 represent alloys with negative $\Delta H$-$T\Delta S_{conf}$, showing that these reactions should proceed spontaneously at the corresponding temperatures and pressures. By comparing Fig. S1 with Fig. S3, high temperature has a positive effect on these reactions, which promotes the alloying processes mentioned above.

Among the elements selected in our simulations, we have selected the non-siderophile elements with moderate and high volatility (Fig. 1). As the pressure increases, more alloys become thermodynamically stable, suggesting some non-siderophile elements gradually tend to form alloys with Fe spontaneously. The shaded areas reflect that the relation between abundance of elements and their thermodynamic stabilities alloying with *hcp*-Fe shows a negative trend, especially at pressures of 150 and 300 GPa. In other words, the binding strengths of these elements alloying with Fe, as quantified by the formation enthalpies, are strongly correlated with their depletion in the silicate Earth. This trend is consistent with the hypothesis that the depletion of these non-siderophile and volatile elements could also be considered by accretion into the inner core as substitutional defects of *hcp*-Fe. Some elements, such as S, Se and Te, have been shown to become more siderophile with increasing pressure[44]. In Fig. 1, the formation enthalpies of alloys Fe$_{127}$S, Fe$_{127}$Se and Fe$_{127}$Te decrease significantly (~ 0.05 eV/atom) as the pressure increases from 20 to 300 GPa. Our simulations are thus in agreement with the previous results[44], which is also a validation of computational scheme of the current study. Our calculations also show that the depletion of several elements, including F, K and Rb, is not likely associated with their assimilation into the Earth's core. Furthermore, we noticed that, with increasing pressure, the formation enthalpies of several alloys, such as Fe$_{127}$Li and Fe$_{127}$B,

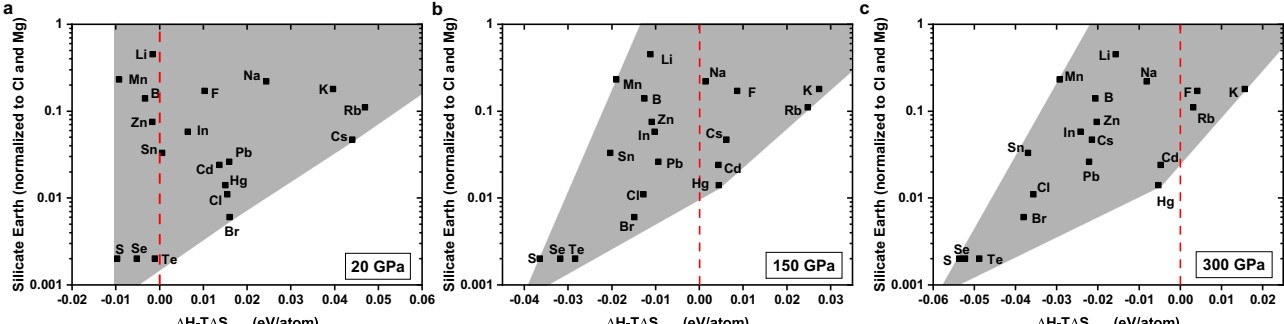

**Fig. 1 | Elemental abundance in the silicate Earth versus stabilities of the Fe alloys at different pressures.** These results were obtained from a simulation cell with a substitution ratio of 1/128 (Fe$_{127}$X) at (**a**) 20 GPa, (**b**) 150 GPa and (**c**) 300 GPa. Elemental abundances in the silicate Earth are ratioed to those in CI carbonaceous chondrites and normalized to $\frac{[Mg]_{Earth}}{[Mg]_{CI}} = 1.0$. The horizontal axes are the terms of $\Delta H - T\Delta S_{conf}$ with different temperatures of 2000, 4000 and 6000 K at 20, 150 and 300 GPa, respectively. The red dashed lines represent the boundaries $\Delta H - T\Delta S_{conf} = 0$ under the corresponding conditions.

decrease slightly (-0.02 eV/atom), indicating that the siderophilicities of these impurity elements will not change significantly during core segregation. It is worth noting that these elements are plotted on the 'volatile trend' line as shown in previous work, which is consistent with the current findings that their depletion could plausibly be ascribed to volatility[24]. Our calculations provide an explanation for the storage of these non-siderophile and volatile elements in the Earth's interior.

## Fe alloys with *p*-block elements

By summarizing the formation enthalpies of these alloys, we found that the Fe alloys with *p*-block elements are typically thermodynamically stable, as shown in Fig. 2. The formation enthalpies of the *p*-block alloys for groups IIIA, IVA, VA and VIA increase with the impurity elements from top to bottom of the periodic table, which can be explained by the electronegativity differences between each element and Fe[45]. As observed in Fig. 2, the formation enthalpy behaviors of the halogens (F, Cl and Br) and noble gases (Ne, Ar, Kr and Xe) decrease with increasing mass, which is related to their high electronegativities under both ambient and high pressure. According to the Hume-Rothery rules, for substitutional solid solutions, the solute and solvent should have similar electronegativity. If there is a large difference of the electronegativity between two elements, the metals tend to form intermetallic compounds instead of solid solutions. For halogens and noble gases, their electronegativities are much higher than Fe at relative low pressures, leading to positive formation enthalpies at 20 and 150 GPa. With increasing pressure, the electronegativity differences between Fe and impurity atoms gradually decreased, and their alloys thus tend to be stable. The lighter halogen atom has a higher

electronegativity, leading to a larger difference with Fe, and the behavior of formation enthalpy for the halogens moves down in the periodic table. Furthermore, due to the higher electronegativity of light halogens and noble gas atoms, they are less likely than the heavy atoms to alloy with Fe, which might offer an explanation for the gradual decrease in formation enthalpies of halogens and noble gases when moving down in the periodic table.

## Electronic properties of Fe-alloys

In order to explore any distinctive electronic features of our proposed Earth core Fe-alloys, Bader charge analyses were conducted to evaluate the charge transfer between Fe atoms and impurity atoms in the systems, as shown in Fig. 3[46]. For most impurities with relatively low ΔH alloys (before Co), the charge of Fe atoms in their alloys remains negative from 20 to 300 GPa, indicating a weaker electronegativity than Fe. The Bader charges of three thermodynamically stable alloys with positively charged iron, $Fe_{127}P$, $Fe_{127}S$ and $Fe_{127}Se$, decrease with the increasing pressure, showing the trend of Fe being an oxidant (electron acceptor). For another two alloys, $Fe_{127}As$ and $Fe_{127}Te$, the charge on Fe changes from positive to negative with increasing pressure, suggesting that the relative electronegativity between Fe and these elements might be switched under pressure and it is more prone to form thermodynamically stable Fe alloys with negatively charged iron at high pressures.

## Wave velocities of Fe alloys

Finally, we take 13 elements of potential alloying with Fe at high pressure as representative systems (P, S, As, Ge, Se, Te, Si, Sb, Ga, Br,

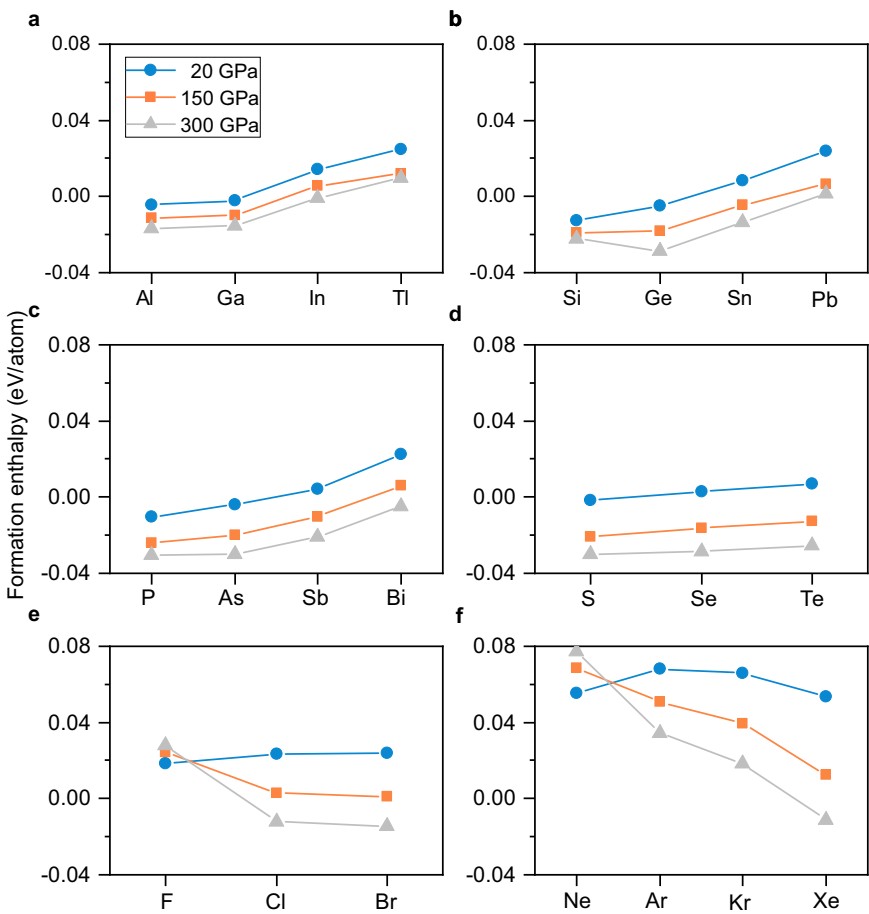

**Fig. 2 | Formation enthalpies of the thermodynamically stable alloys $Fe_{127}X$.** Each element in the horizontal coordinate represents X in $Fe_{127}X$. Panels (**a**–**f**) represent groups IIIA, IVA, V, VIA, VIIA and VIIIA in the periodical table, respectively.

The simulations are performed at 20, 150 and 300 GPa, with symbols blue circles, orange squares and gray triangles, respectively. Formation enthalpy is calculated by using formula (1).

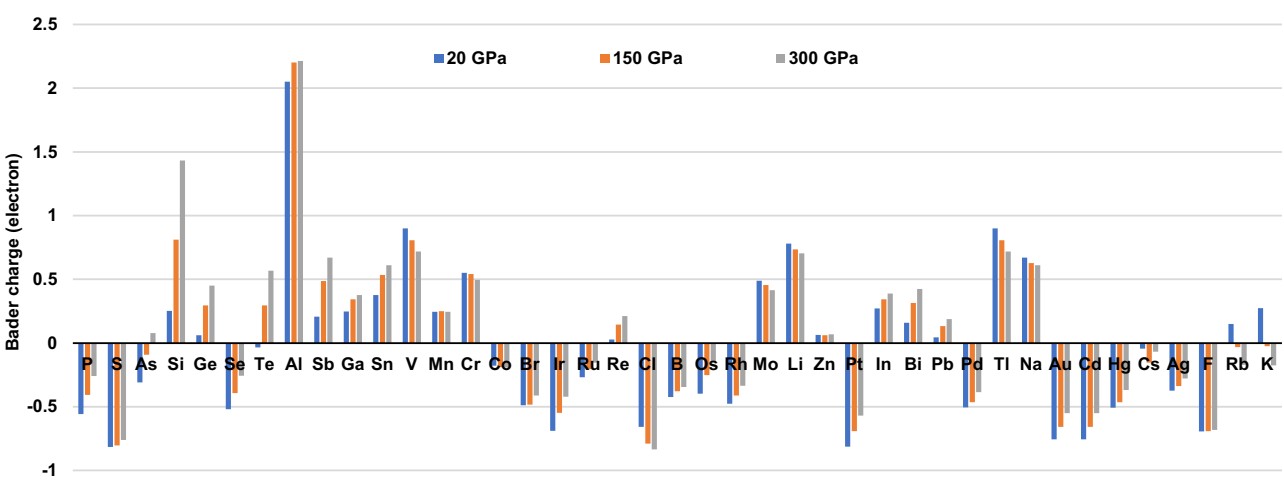

**Fig. 3 | Calculated Bader charge of the depleting element in Fe alloys.** These simulations were performed with a substitution ratio of 1/128 (Fe$_{127}$X) at 20, 150 and 300 GPa, with color blue, orange and gray, respectively. Here, positive values of Bader charge represent Fe atoms gain electrons, and negative values represent Fe atoms lose electrons. The elements are arranged in order of the formation enthalpies of alloys at 150 GPa.

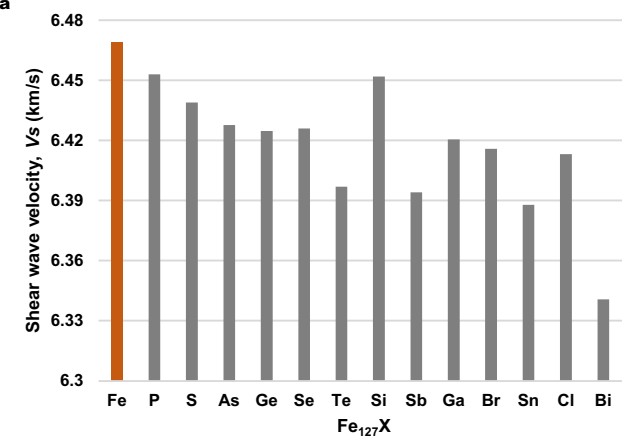

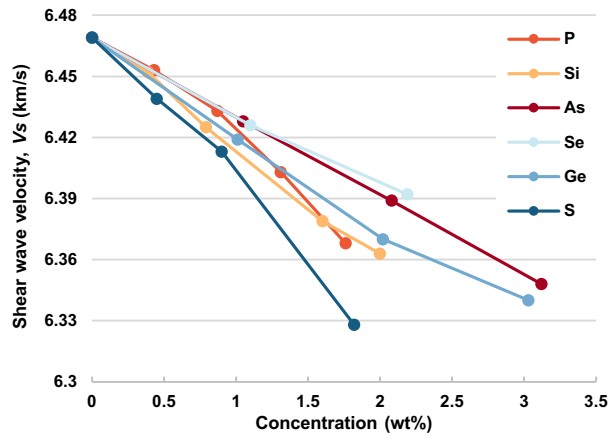

**Fig. 4 | Static shear wave velocities ($V_S$) of Fe alloys at 300 GPa. a** Comparison of different doping elements with a substitutional ratio of 1/128. Each element in the horizontal coordinate represents X in Fe$_{127}$X. **b** Effects of doping concentrations for different elements on $V_S$. Different doping ratios are accomplished by substituting different numbers of atoms randomly in *hcp*-Fe supercells.

Sn, Cl and Bi) to investigate the elastic constants of the expected structures (Table S4). These elements are selected from Fig. S3, which includes a wide range of elements. The Fe alloys with these elements have the lowest formation enthalpies at 300 GPa, as shown in Table S3, indicating that they could be potentially stored in the Earth's inner core, based on our results. According to the density deficit between pure Fe and PREM (-10% at outer core and 3.6% at inner core boundary)[47–49], the doping proportion of elements with heavier relative atomic mass than Fe (heavy-elements) should be low, so that high density should be compensated without the need for a large concentration of elements lighter than Fe (light-elements). We first calculated the elastic constants of the model Fe$_{127}$X structures and compared computed values with those of pure Fe. The calculated shear wave velocities $V_S$, primary wave velocities $V_P$, bulk wave velocities $V_\Phi$ and Possion's ratios are shown in Fig. 4a, Fig. S4, S5 and S6, respectively. Intriguingly, the results show that the incorporation of different elements will reduce the $V_S$ of pure Fe, but to significantly different degrees. We attribute this result to a slight lattice distortion of *hcp*-Fe after substituting Fe with other atoms. Lattice distortion could reduce the shear modulus of the alloys (Fig. S7), which further reduces $V_S$ according to equation 4. Thus, owing to this effect, $V_S$ will be reduced regardless of which elements are doped. For 'heavy'

impurities, the effect of density will aggravate this reduction of $V_S$, converse to typical expectations. To evaluate the doping effect on shear wave velocities of these alloys, we have calculated $V_S$ of Fe alloys with either light-elements or heavy-elements up to about 2 wt% (Fig. 4b). As the doping concentration increases, the $V_S$ of the alloys linearly decreases. The structures of these model systems with different concentrations are shown in Fig. S8.

## Discussion

The densities, wave velocities and compressibility of iron-rich alloys under high P–T conditions, compared with observed seismic models, such as PREM, serves as an important constraint on the Earth's core composition[27]. Previous studies have primarily focused on 'light' elements, as candidate impurities of the Fe alloys[24]. Our results suggest that certain light and heavy elements, which are non-siderophile and volatile, could be potentially become captured into the Earth's core during metal–silicate segregation, to enrich the chemical composition of the core. Based on the relationship between static wave velocities and high temperature $V_S$ in the systems Fe-C, Fe-O and Fe-Si reported previously[20,47], the static $V_S$ for the elements shown in Fig. 4 could be extrapolated to high temperatures. The $S$-wave velocities in the inner core observed from seismology (-3.7 km/s) are lower than pure *hcp*-Fe

obtained from experimental and theoretical results (~4.5 km/s)[27]. To extrapolate our static findings to elevated temperatures, we compared the calculated $V_S$ of pure $hcp$-Fe with that of Fe alloys in previous works at high temperatures and 0 K in Fig. S9. The data from previous studies (dashed lines) suggest that 'light' atoms are likely to soften the static $V_S$ of $hcp$-Fe, and this effect remains similar at elevated temperatures. This trend indicates that the doping of 'heavy' elements we have considered at 0 K (red open squares) in Fig. 4a may have the potential to decrease $V_S$ of $hcp$-Fe at high temperatures. As doping concentrations increase, the static $V_S$ of the alloys linearly decreases (Fig. 4b), suggesting that high temperature $V_S$ of Fe alloys may be decreased due to doping of these heavier impurity atoms. Our current findings thus suggest the estimation of the concentration of 'light' elements within the Earth's mantle as well as the Earth's core remain an open challenge.

In conclusion, we have performed computational simulations on the possible incorporation of a range of elements into the Earth's core, by considering the thermodynamic stabilities of corresponding Fe alloys. At high pressure, many elements, that are typically assumed lithophile or chalcophile, may combine with Fe, with the formation enthalpies of these Fe alloys decreasing with increasing pressure. Among these alloys, we found that the abundances of non-siderophile and volatile elements are positively correlated with the formation enthalpies of these Fe alloys at high pressures, consistent with the inference that depleted non-siderophile elements might be stored in the core as impurities substituted in the $hcp$-Fe. In addition, by analyzing the electronic properties of the range of alloys considered here, we noticed that in those with relatively low formation enthalpies, Fe tends to be an oxidant at pressures above 20 GPa. Furthermore, our static wave velocity calculations show that both light and heavy elements have a weakening effect compared to that of pure Fe, due to small lattice distortions upon substitution, and this effect intensifies with increasing doping concentration.

## Methods

In this work, we performed *ab* initio computational simulations with models containing up to 128 atoms (a $4 \times 4 \times 4$ $hcp$ supercell), in which 1-6 impurity atoms are substituted into the supercell (Fig. S8). These models with different impurity concentrations were used to simulate the Fe alloys during the formation of the Earth's core. The impurities were chosen to take substitutional sites instead of interstitial sites due to the relatively large atomic radii of these elements (the list of elements and their atomic radii are shown in Table S5). In order to verify the stability of the substitutional model structures at high temperatures, we conducted molecular dynamics simulations on the substitutional and interstitial models ($Fe_{127}As_1$, $Fe_{128}As_1$, $Fe_{124}As_4$, and $Fe_{128}As_4$) under Earth's inner core conditions (333 GPa for $Fe_{127}As$, 340 GPa for $Fe_{124}As_4$, 345 GPa for $Fe_{128}As$ and 330 GPa for $Fe_{128}As_4$ at 6000 K), as shown in Fig. S10. In these simulations, there is a diffusion behavior for Fe atoms in the interstitial alloy model, which indicates a liquid state of the system and substitutional alloy structures are thus much better in further simulations for the understanding of Earth's inner core. The simulations on the structure geometry optimization are based on the framework of density functional theory[48,49]. The exchange correlation function was chosen as Generalized Gradient Approximation Perdew-Burke-Ernzerhof [50,51] by employing projector-augmented wave method[52] as implemented in the VASP code[53]. We choose energy cutoff of 1000 eV and $k$-point sampling of $2\pi \times 0.03$ Å$^{-1}$ in the Brillouin zone[54,55]. We have performed additional simulations to explore any influence of the spin-polarized effect for Fe. Our results show that there is no spin polarization for Fe over a pressure range of 20–300 GPa (Table S2). To estimate the relative thermodynamic stabilities, we employed the $hcp$ supercell of $Fe_{127}X$ (X represents the defective elements as shown in Fig. S8a) at 20, 150 and 300 GPa and

0 K with the following formula:

$$\Delta H = \frac{127 \times H(\text{Fe}) + H(X) - H(\text{Fe}_{127}X)}{128} \qquad (1)$$

where $\Delta H$ is the formation enthalpy per atom and $H$ is the calculated enthalpy per chemical unit for each compound. For defective systems, the configurational entropy $S_{conf}$ is estimated by the following formula:

$$S_{conf} = -k_B[m\ln(m) + (1-m)\ln(1-m)] \qquad (2)$$

where $k_B$ is the Boltzmann constant ($1.380649 \times 10^{-23}$ J/K) and m is the defect concentration.

The elastic constants were calculated using the stress–strain method. Voigt average scheme where the strain is taken to be uniform was carried out to determine the shear modulus (G), bulk modulus (B) and density ($\rho$) of the alloys, which proved more appropriate and accurate in calculating the seismic wave properties.

The primary wave velocity $V_P$, shear wave velocity $V_S$ and bulk wave velocity $V_\Phi$ are:

$$V_P = \sqrt{\frac{B + \frac{4G}{3}}{\rho}}, V_S = \sqrt{\frac{G}{\rho}}, V_\phi = \sqrt{\frac{B}{\rho}} \qquad (3-5)$$

Possion's ratio was calculated from:

$$\nu = \frac{3B - G}{2(3B + G)} \qquad (6)$$

## Data availability

The main data generated in this study are provided in the Supplementary Information, Source Data file as well as Figshare. Source data are provided with this paper.

## Code availability

The code for first-principles-related simulations is a commercial code and can be found at https://www.vasp.at/ (VASP).

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

## Acknowledgements

This work was supported by the National Key Research and Development Program of China (Grant No. 2022YFA1402304), National Natural Science Foundation of China (Grant No. 12074138, 52288102, and 52090024), Program for Jilin University Science and Technology Innovative Research Team (2021TD–05), Jilin Province Science and Technology Development Program (Grant No. YDZJ202102CXJD016), the Program for Jilin University Computational Interdisciplinary Innovative Platform, the Fundamental Research Funds for the Central Universities and computing facilities at the High-Performance Computing Center of Jilin University. The authors thank Prof. Yanming Ma for valuable discussions.

## Author contributions

H.L. designed the research. Y.T. performed theoretical work. W.Z. and X.F. contributed comments. Y.T., P.Z., S.A.T.R and H.L. wrote the paper. All authors contributed to discussing the results and writing the paper.

## Competing interests

The authors declare no competing interests.
