## [Peer Review File · Nature Communications]

REVIEWER COMMENTS

Reviewer #1 (Remarks to the Author):

This paper provides new results regarding potentially incorporating a range of elements in the Earth's core. These conclusions are drawn from computational simulations that assess the thermodynamic stability of forming iron (Fe) alloys at core conditions. The data presented here represents a step forward in comprehending how elements behave under Earth's core conditions. This information has the potential to be valuable for research in both geology and geophysics.

However, several things need to be improved to clarify and support the conclusions reached by the authors before this work can be published. The paper is confusing, and more detailed information on several aspects is needed.

A significant issue I've identified pertains to the introductory section, where it is stated that the authors will primarily concentrate on elements that are notably scarce in the Earth's mantle and are categorized as non-siderophile volatiles. However, the research encompasses a wide array of elements, including Se, Te, and S, which have been demonstrated to be highly siderophile elements under core-forming conditions (Rose-Weston et al., *Geochim. Cosmochim. Acta* 73, 4598–4615, 2009). Additionally, there is a lack of clarity concerning what elements are being considered as volatiles. Are they elements with lower condensation temperatures (temperature by which 50% of the element would have condensed from a gas of solar composition at a total pressure of 10^{-4} bar (Lodders, 2003)).? If so, why are elements like Tl or Ge not shown in Fig. 1? And what specific threshold is being applied for their inclusion? Is the selection based solely on elements that deviate from the trend in volatility when comparing their abundances in the silicate Earth to their condensation temperatures? Providing a comprehensive list of all the elements under study and elucidating the criteria for their inclusion is primordial for the clarity of this work.

One of the paper's key findings is the positive correlation between the abundances of elements classified as non-siderophile and volatile (although it's worth noting that not all of these elements fit this classification) and the formation enthalpies of Fe alloys under high pressures. This correlation suggests that these elements are incorporated into the Earth's core. However, upon closer examination of Figure 1, it appears that some elements are not represented in the figure and are only available in supplementary data. This raises the concern that the positive correlation observed might be due to an arbitrary selection of elements in Figure 1, and more elements could affect this relationship than initially shown. Thus, again, it is pivotal to clarify the selection criteria for the elements in question to see if the correlation is real.

It would be valuable if the authors could enhance their argument by comparing their results with existing experimental and theoretical data on partition coefficients for certain elements under core-forming conditions. This would add strength to their findings.

Finally, the connection between static shear wave velocities (V_s), the concentration of different doping elements in the Fe alloys, and their effect on approaching PREM needs more work. Currently, this correlation is mentioned but hasn't been thoroughly examined or discussed.

More specific comments can be found in the attached files.

Reviewer #2 (Remarks to the Author):

Overall thoughts:

This was a large, comprehensive study examining the thermodynamic stability of substitutional iron alloys. The work is novel in that it is the first first-principles study to examine such a wide array of elements being incorporated into the hcp-Fe structure with this substitution mechanism. The amount of data is commendable, the methods are sound, and the work will have particular interest for the inner core community. However, the paper is missing a key discussion of the role of liquid iron in their story, particularly during core-mantle differentiation when the entire planet was molten. In general, the paper could use more detailed discussion of the implications of their results for the evolution of the Earth as a whole. More detailed comments are attached below.

Summary: Certain volatile and non-siderophile elements, such as Se and Br, are over-depleted in the silicate Earth compared to the established volatility trend. In this work, the authors perform first-principles simulations examining whether these elements can be incorporated into the Earth's core as substitutional alloys. They find that these alloys are thermodynamically stable, highlighting a potential reservoir for volatile elements within the Earth's deep interior.

Comments:

- 1) The introduction would be improved by including a quick overview of previous work in the field (both experimental and computational) on iron alloys in the inner core.
- 2) In the introduction, I agree that the positive correlation of the formation enthalpies of the Fe-alloys with the abundances of the non-siderophile and volatile elements indicates that these elements could be sequestered in the solid inner core today. However, what seems to be missing is the role that liquid iron plays in this story. Since inner core crystallization occurred after differentiation, the abundance of non-siderophile and volatile elements in the bulk silicate Earth would be primarily set by the partitioning of these elements between silicate and liquid metal. Incorporation of these elements as substitutional alloys in the inner core would occur later, affecting their abundances in the liquid outer core. Drawing a direct link between these two processes is confusing.
- 3) Were these calculations spin-polarized? Please indicate in the methods section.
- 4) For easier comparison, it would be helpful to add a dotted line at 0 in Figure S3 (like you did in Figure S4), and to plot the points from Figure S3 on Figure S4 (faded, or in a different color).
- 5) Please cite which geotherm you are using in line 121.
- 6) In Figure S5, what do you mean by "the fitting of the data?" What exactly was fit? The meaning of the dashed red line is unclear.
- 7) In general, the figures are quite small and their resolution could be improved.
- 8) While the trend in figure 1 is evident, especially at higher pressures, the spread between different elements is quite high, indicating other factors are coming into play. Adding possible explanations for outliers in the expected trend (such as Li, Mg, B) would be helpful here in your discussion of the figure.
- 9) For comparison purposes in Figure 2, make the y axis scale the same for all plots.
- 10) Do you have an explanation for the different formation enthalpy behavior of the halogens as you move down the periodic table? They are not included in the discussion of Figure 2.
- 11) The discussion would benefit from an additional paragraph at the end, explaining in more detail the implications of your results for the thermochemical evolution of the Earth. For example, discussion of

the expected composition of the inner core, outer core, and mantle from your results and the effect this has on physical and transport properties.

Reviewer #1 (Remarks to the Author):

Response to the reviewer #1

Comments from the Reviewer #1: *This paper provides new results regarding potentially incorporating a range of elements in the Earth's core. These conclusions are drawn from computational simulations that assess the thermodynamic stability of forming iron (Fe) alloys at core conditions. The data presented here represents a step forward in comprehending how elements behave under Earth's core conditions. This information has the potential to be valuable for research in both geology and geophysics.*

However, several things need to be improved to clarify and support the conclusions reached by the authors before this work can be published. The paper is confusing, and more detailed information on several aspects is needed.

Response: We are grateful to the reviewer for a careful reading, the positive comments and constructive suggestions, with which the quality of the work has been improved greatly. According to the comments and suggestions, we have revised our manuscript in the attached files point by point.

Comments from the Reviewer #1: *A significant issue I've identified pertains to the introductory section, where it is stated that the authors will primarily concentrate on elements that are notably scarce in the Earth's mantle and are categorized as non-siderophile volatiles. However, the research encompasses a wide array of elements, including Se, Te, and S, which have been demonstrated to be highly siderophile elements under core-forming conditions (Rose-Weston et al., Geochim. Cosmochim. Acta 73, 4598–4615, 2009). Additionally, there is a lack of clarity concerning what elements are being considered as volatiles. Are they elements with lower condensation temperatures (temperature by which 50% of the element would have condensed from a gas of solar composition at a total pressure of 10^{-4} bar (Lodders, 2003)).? If so, why are elements like Tl or Ge not shown in Fig. 1? And what specific threshold is being applied for their inclusion? Is the selection based solely on elements that deviate from the trend in volatility when comparing their abundances in the silicate Earth to their condensation temperatures? Providing a comprehensive list of all the elements under study and elucidating the criteria for their inclusion is primordial for the clarity of this work.*

One of the paper's key findings is the positive correlation between the abundances of elements classified as non-siderophile and volatile (although it's worth noting that not all of these elements fit this classification) and the formation enthalpies of Fe alloys under high pressures. This correlation suggests that these elements are incorporated into the Earth's core. However, upon closer examination of Figure 1, it appears that some elements are not represented in the figure and are only available in supplementary data. This raises the concern that the positive correlation observed might be due to an arbitrary selection of elements in Figure 1, and more elements could affect this relationship than initially shown. Thus, again, it is pivotal to clarify the selection criteria for the elements in question to see if the correlation is real.

Response: We thank the reviewer for their comments on the criteria for the selection of the elements as shown in Fig. 1. In fact, the elements we chose in Fig. 1 were not arbitrary, but relied on their 50% condensation temperatures and siderophilicities. In the current work, the classification of the elements in Fig. 1 are referred as to a review paper {W.F. McDonough *et al.*, *Chemical Geology* 120, 223 (1995)}. For convenience, here we show the classification of the elements as mentioned in that review reference in Table R1 below.

Table R1. Classification of the elements that we have considered in this work, including main text and supplementary information. Volatility of elements is classified according to 50% condensation temperatures (T_c) at 10^{-4} bar.

	Lithophile	Siderophile	Chalcophile
Refractory ($T_c \geq 1400$ K)	Al, V	Mo, Ru, Rh, W, Re, Os, Ir, Pt	
Transitional (~ 1350 K $> T_c > \sim 1250$ K)	Mg, Si, Cr	Fe, Co, Ni, Pd	
Moderately volatile (~ 1250 K $> T_c > \sim 800$ K)	Li, B, Na, K, Mn, Rb, Cs	P, Cu, Ga, Ge, As, Ag, Sb, Au	
Highly volatile ($T_c < 800$ K)	F, Cl, Br, Zn	Tl, Bi	S, Se, Cd, In, Sn, Te, Hg, Pb

In the current manuscript, we focused more on lithophile and chalcophile elements (non-siderophile elements as noted in the text) with moderate and high volatility, which are marked in green in Table R1. On one hand, these elements were employed to investigate their formation enthalpies upon alloying into *hcp*-Fe at high pressures, since it was accepted that other kinds of elements (the siderophile elements or elements with low volatility) are likely to be retained in Earth's iron core during its segregation [Kei Hirose *et al.*, *Light elements in the Earth's core. Nat. Rev. Earth Environ.* 2, 645–658 (2021)]. On the other hand, it is known that pressure is a unusual tool to uncover new physics or chemistry that is inaccessible at ambient pressure, which allow us to investigate if some lithophile and chalcophiles elements could become kind of siderophile elements under high pressures. Based on this criterion as mentioned above, elements Se, Te and S are included in the Fig. 1, because they are not siderophile, but without Tl or Ge due to siderophile property. Besides, we have also included more elements that are siderophile or with condensation temperatures above 1250 K to investigate the thermodynamic stabilities of their Fe alloys under high pressures, as shown Fig. S5. As a result of these simulations, we indeed found that a great number of elements tend to be incorporated into iron lattice under high pressure. In response, we have added detailed information on the criteria for the inclusion of the elements in Fig. 1 in the revised manuscript.

“As has been discussed previously, the elements that appear depleted in the silicate Earth can be ordered in terms of their 50% condensation temperatures at 10^{-4} bar, and in terms of their siderophilicities (Table S2). Here, we focus on the non-siderophile elements (noted as lithophile and chalcophile in Table S2) with moderate and high volatility (shown in green in Table S2), which are represented in Figure 1. We have investigated their formation enthalpies upon

alloying into *hcp*-Fe at high pressures, since the depleted elements with high siderophilicity or low volatility can be presumed to have been sequestered within the deep Earth during the core segregation. Furthermore, we have also included additional elements as shown in Table S2 (Figure S1), to investigate their possible alloying propensity with Fe at high pressures.”

Comments from the Reviewer #1: *It would be valuable if the authors could enhance their argument by comparing their results with existing experimental and theoretical data on partition coefficients for certain elements under core-forming conditions. This would add strength to their findings.*

Response: We thank the reviewer for the good suggestions. In response, we compared our results with the previous works for S, Se and Te, and added the pertinent discussion in the revised manuscript.

“Some elements, such as S, Se and Te, are seen to become more siderophile with increasing pressure.^{46,47} In Figure 1, the formation enthalpies of alloys Fe₁₂₇S, Fe₁₂₇Se and Fe₁₂₇Te decrease significantly (~ 0.05 eV/atom) as the pressure increases from 20 GPa to 300 GPa. Our simulations are thus in agreement with the previous results, which is also a validation of computational scheme of the current study.”

[46] Lesley Rose-Weston, James M. Brenan, Yingwei Fei, Richard A. Secco and Daniel J. Frost, Effect of pressure, temperature, and oxygen fugacity on the metal-silicate partitioning of Te, Se, and S: Implications for earth differentiation, *Geochimica et Cosmochimica Acta* 73, 4598 (2009).

[47] Zaicong Wang and Harry Becker, Ratios of S, Se and Te in the silicate Earth require a volatile-rich late veneer, *Nature* 499, 328–331 (2013).

Comments from the Reviewer #1: *Finally, the connection between static shear wave velocities (V_s), the concentration of different doping elements in the Fe alloys, and their effect on approaching PREM needs more work. Currently, this correlation is mentioned but hasn't been thoroughly examined or discussed.*

Response: We thank the reviewer for the good suggestions. Estimation of V_s in Fig. 4b is difficult to be compared with PREM value directly, owing to our static calculations. In order to extend the results from 0 K to high temperatures, we compared the relationship between static wave velocities and high temperature V_s in the systems Fe-C, Fe-O and Fe-Si from previous studies {Martorell, B., Wood, I. G., Brodholt, J. & Vočadlo, L. The elastic properties of *hcp*-Fe_{1-x}Si_x at Earth's inner-core conditions. *Earth Planet. Sci. Lett.* 451, 89–96 (2016); Yu He, Shichuan Sun, Duck Young Kim, Bo Gyu Jang, Heping Li and Ho-kwang Mao, Superionic iron alloys and their seismic velocities in Earth's inner core, *Nature* 602, 258 (2022)}, and extrapolated the trend to the elements as mentioned in Fig. 4. The analysis contributes to understanding the connection between static shear wave velocities (V_s), the concentration of different doping elements in the Fe alloys, and their effect on approaching PREM. In response, we have added pertinent discussion in the revised manuscript.

“The *S*-wave velocities in the inner core observed from seismology (~3.7 km/s) are lower than

pure *hcp*-Fe obtained from experimental and theoretical results (~ 4.5 km/s).²⁷ To extrapolate our static findings to elevated temperatures, we compared the calculated V_s of pure *hcp*-Fe with that of Fe alloys in previous works at high temperatures and 0 K in Figure S9. The data from previous studies (dashed lines) suggest that ‘light’ atoms are likely to soften the static V_s of *hcp*-Fe, and this effect remains similar at elevated temperatures. This trend indicates that the doping of ‘heavy’ elements we have considered at 0 K (red open squares) in Figure 4a may have the potential to decrease V_s of *hcp*-Fe at high temperatures. As doping concentrations increase, the static V_s of the alloys linearly decreases (Figure 4b), suggesting that high temperature V_s of Fe alloys may be decreased due to doping of these heavier impurity atoms.”

Figure R1. Calculated shear wave velocity (V_s) for *hcp*-Fe, $\text{FeC}_{0.0625}$, $\text{FeO}_{0.0625}$, $\text{Fe}_{0.9375}\text{Si}_{0.0625}$ at 0 K and high temperatures. Black solid line represents pure Fe, dash lines represent Fe alloys in previous works and red open squares represent the alloys in Figure 4b.

Comments from the Reviewer #1: *More specific comments can be found in the attached files.*

Response: We thank the reviewer for the good suggestions. Kindly find below our detailed response to all issues point by point.

1) *I will suggest deep interior as these findings also would affect the mantle composition*

Response: We have updated the **Abstract** section in the revision as follows:

“We suggest potential reservoirs for volatile elements within the deep Earth, augmenting our understanding of the deep Earth’s composition.”

2) *Why do you mention only these elements and not S and Te, for example, which are also considered volatile?*

Response: The two elements mentioned here “Se and Br” are two examples of volatile elements. For S, Te and other volatile elements, a more detailed study has been presented in Fig. 1. Here, we have rephrased the sentence in the revised manuscript.

“However, by comparing the abundance of the elements with their approximate position in the trend line, some elements appear excessively depleted in the silicate Earth, such as Se and Br,

suggesting depletion could be attributed to a combination of both volatility and dissolution or assimilation into the core.”

- 3) *I imagine you are referring to the depletion of volatile elements. Please rephrase as this text is not very clear. What do you refer to when mentioning solid Earth? Is it the silicate Earth?*

Response: We are thankful for the reviewer’s constructive suggestion. In the revised abstract, we have rewritten the sentence:

“It is noted that in most scenarios associated with depletion due to assimilation in the core, the depleted elements are expected to partition into the metallic (iron) core as a low solute concentration of iron.”

- 4) *It has been shown that Se and Te apart from being volatile also behave as highly siderophile elements at core forming conditions (Rose-Weston, L., Brenan, J. M., Fei, Y., Secco, R. A. & Frost, D. J. Effect of pressure, temperature, and oxygen fugacity on the metal–silicate partitioning of Te, Se, and S: implications for Earth differentiation. Geochim. Cosmochim. Acta 73, 4598–4615 (2009).) For S, it has been shown that it behaves as moderately siderophile (e.g., Labidi, J., et al., Nature 501, 208–211 (2013))*

Response: As we explained above, elements like S, Se and Te are considered as chalcophile elements, which are also noted as non-siderophile in the text (the classification is shown in revised Supplementary Information). In order to make the results more systematic, we included all the non-siderophile and volatile elements within the list of our study, even if S, Se and Te have been demonstrated to be more siderophile with the increasing pressure. Our simulations are thus not in contradiction with the previous results, which is also a validation of computational scheme as simulated in the current study.

- 5) *Where? In the mantle, core, silicate earth? Please be more specific*

Response: Here, we mean that at the inner core pressure.

- 6) *Here you could expand a little bit more as at the moment is quite disconnected from the previous text*

Response: We have added the following sentence in the **Introduction** section.

“It is generally supposed that the low shear velocities (V_S) observed in seismic data from the core imply an inner core density lower than that of pure iron. This suggests the existence of ‘light’ elements as the major alloying candidates within it, such as H, C, O, S and Si.”

- 7) *It will be good to state briefly that this was done to simulate the formation of Fe alloys in the Earth’s core.*

Response: We have added the following statement in the **Method** section of the revised manuscript:

“These models with different impurity concentrations were used to simulate the Fe alloys during the formation of the Earth’s core.”

8) *Which elements are these exactly? Please provide that information so we can easily check data such as the atomic radii without trying to figure out from figures only*

Response: We have added the list of the elements and their atomic radii as Table S1 in the Supplementary Information.

9) *This doubt may arise mainly because I am unfamiliar with these computation simulations. Still, I don't understand why your ab initio computational simulations are done with 128 atoms ($Fe_{128-n}X_n$), but when you conduct the molecular dynamic simulations, you do it with models with 144 and 148 ($Fe_{140}X_4$ and $Fe_{144}X_4$). Please clarify.*

Response: In fact, $Fe_{140}X_4$ and $Fe_{144}X_4$ were created by the actual $2 \times 2 \times 1$ supercells of $Fe_{35}X_1$ and $Fe_{36}X_1$, which were employed to study the possibility of substitutional and interstitial Fe alloy with an element, respectively. Because we would study the effect with doping different number of impurity atoms into Fe alloy, to create a $2 \times 2 \times 1$ supercell of 36 atoms (the simulated cell contains 144 atoms) is much more convenient to randomly distribute the doping elements within the cell. We have added these details in the revised **Supplementary Information** (Figure S11) and the structures of $Fe_{140}X_4$ and $Fe_{144}X_4$ in Figure S1. In response, we have also performed additional molecular dynamics simulations on $Fe_{127}X_1$ and $Fe_{124}X_4$, and the results show the same conclusion as simulated in the supercell of $Fe_{140}X_4$ and $Fe_{144}X_4$.

“In order to verify the stability of the substitutional model structures at high temperatures, we conducted molecular dynamics simulations on the substitutional and interstitial models ($Fe_{127}As_1$, $Fe_{128}As_1$, $Fe_{124}As_4$, and $Fe_{128}As_4$) under Earth's inner core conditions (333 GPa for $Fe_{127}As$, 340 GPa for $Fe_{124}As_4$, 345 GPa for $Fe_{128}As$ and 330 GPa for $Fe_{128}As_4$ at 6000 K), as shown in Figure S10. In these simulations, there is a diffusion behavior for Fe atoms in the interstitial alloy model, which indicates a liquid state of the system and substitutional alloy structures are thus much better in further simulations for the understanding of Earth's inner core.”

10) *In Fig. S2 you use As as the impurity and thus as the volatile element in question. However, As has a temperature of 50% condensation similar to HSE (1065 K so moderately volatile), so it is not really considered a volatile element. I would like to see the same simulation with the volatile elements mentioned in this paper, eg. Se or Br.*

Response: In this work, the MD simulations shown in Fig. S2 were performed to not study whether the element is volatile or not, but to compare the two doping models (substitutional and interstitial). The MSDs in interstitial model were found to increase obviously with simulation time, indicating a liquid state, suggesting that the interstitial models are not suitable for the traditional understanding under the IC conditions. Therefore, we chose substitutional model for Fe alloys in our following simulations. Moreover, we also found that this kind of simulation is also used in the previous work (Yu He et al., Nature, 602, 258 (2022)). In response to the suggestions of this reviewer, we have performed the additional MD simulations for the Fe-Se system with two alloy models as shown in Fig. S10.

We have clarified this point in the revision.

“In these simulations, there is a diffusion behavior for Fe atoms in the interstitial alloy model, which indicates a liquid state of the system and substitutional alloy structures are thus much better in further simulations for the understanding of the inner core (Figure S10).”

11) 332 GPa for interstitial model and 335 GPa for substitutional model as stated in Fig. S2

Response: We are grateful to this reviewer for a very careful reading. This error has been corrected in the revised manuscript.

12) You should provide the value here and also always state all the values and constant used in your equations.

Response: We have added the values of k_B in the revised manuscript.

13) State in the text that ρ refers to density.

Response: We have added the statement about ρ in the revised manuscript.

14) These 40 elements are not all volatiles. Again, it is important to clarify why you selected these elements specifically. Based on what values (e.g., condensation temperature)?

Response: Related description has been responded above. We have strengthened the description of the selection rules in the revised **Result** section, adding the following discussions.

“As has been discussed previously, the elements that appear depleted in the silicate Earth can be ordered in terms of their 50% condensation temperatures at 10^{-4} bar, and in terms of their siderophilicities (Table S2). Here, we focus on the non-siderophile elements (noted as lithophile and chalcophile in Table S2) with moderate and high volatility (shown in green in Table S2), which are represented in Figure 1. We have investigated their formation enthalpies upon alloying into *hcp*-Fe at high pressures, since the depleted elements with high siderophilicity or low volatility can be presumed to have been sequestered within the deep Earth during the core segregation. Furthermore, we have also included additional elements as shown in Table S2 (Figure S1), to investigate their possible alloying propensity with Fe at high pressures.”

15) Results are also shown at 20 GPa, so not only core conditions. Also, as part of the supplementary information, please provide the list of the 40 elements selected for the enthalpy calculations so it is easier to check besides the info in the figure.

Response: We have changed the “core conditions” to “20, 150 and 300 GPa” in this sentence. And we also added the list of the 40 selected elements and their calculated enthalpies in revised Supplementary Information as Table S3.

16) Several elements shown in Fig 1 are considered volatile siderophiles (see Fig. 1 of Wood et al., *American Mineralogist* 2019; 104 (6): 844–856). Also, why didn't you select Br, which you mention even in the abstract of this work (not shown in Figure 1)? Please clarify the selection criteria for these elements as below, you mention that you observe a negative trend of the stabilities and alloys, but that is heavily dependent on the selected elements in the plot. Also, clarify the selection criteria regarding volatility (is it condensation T° ?).

Response: We thank the review for the careful reading. The selection criteria have been clarified in the first paragraph of the **Result** section in the revised manuscript. For Br, as a volatile and non-siderophile element, we indeed omitted it in Figure 1 of the previous manuscript through an oversight. And we have corrected Figure 1 in the revision. Also, we have checked the elements involved in Figure 1 again based on the classification of the elements in

the Table S2.

17) *Not clear why other elements like Br, As or Ge are not shown here (all considered volatile siderophile as Se, Te and S).*

Response: Br has been added to the modified Figure 1 as stated in the response to the previous comment. Based on the selection criteria mentioned above, Se, Te and S belong to chalcophile elements (Table S2), but As and Ge are siderophile, thus As or Ge are not shown in Figure 1.

18) *It is worth to mention here that these elements plot on the volatile trend (Fig. 1, Wood et al., 2019). So it is consistent with your findings.*

Response: We thank the reviewer for this suggestion. We have added the following discussions. “It is worth noting that these elements are plotted on the ‘volatile trend’ line as shown in previous work, which is consistent with our findings that their depletion could plausibly be ascribed to volatility.”

19) *No Br is shown here; however, in all supplementary figures, it has been shown and is also part of your discussion. If that is not the case, please explain why you choose not to plot it here.*

Response: We thank this reviewer again for the reminder of Br element in Fig. 1.

20) *Please provide the same y axes in the figure, as it allows for a clearer comparison of the elements between the different groups. This also should be something to improve in your supplementary figures.*

Response: We thank this reviewer for his/her suggestion. In response, we have updated the Figure 2 with the same y axes. We also adjusted the figures in Supplementary Information.

21) *Except for Groups VII and VIII*

Response: We thank this reviewer for careful reading. We have corrected the sentence in the manuscript as follows:

“The formation enthalpies of the p-block alloys for groups IIIA, IVA, VA and VIA increase with the impurity elements from top to bottom of the periodic table, which can be explained by the electronegativity differences between each element and Fe”.

22) *What about Cl and Br, which are not noble gases? Also, what do you refer to when the noble gas elements' electronegativity increases? The values for these elements are not higher than others like in Group VI, for example (see Fig. 1 Tantardini C, Oganov AR. Thermochemical electronegativities of the elements. Nat Commun. 2021).*

Response: The electronegativity of the elements changes with pressure (as mentioned in Fig. 4, Xiao Dong, et al., Electronegativity and chemical hardness of elements under pressure. Proc. Natl. Acad. Sci. U.S.A. 119, 10 (2022)). According to the Hume-Rothery rules, for substitutional solid solutions, the solute and solvent should have similar electronegativity. If the electronegativity difference is large, the metals tend to form intermetallic compounds instead of solid solutions. For halogens (F, Cl and Br) and noble gases (Ne, Ar, Kr and Xe), their electronegativities are much higher than Fe at relative low pressures, leading to positive

formation enthalpies at 20 and 150 GPa. With the pressure increased, the electronegativity differences between Fe and these impurity atoms gradually decreased, so the alloys tend to be stable at 300 GPa.

The expression in the manuscript 'noble gas elements' electronegativity increases' was incorrect. We have corrected mistakes and rewritten the discussion part as follows:

“As observed in Figure. 2, the formation enthalpy behaviors of the halogens (F, Cl and Br) and noble gases (Ne, Ar, Kr and Xe) decrease with increasing mass, which is related to their high electronegativities under both ambient and high pressure. According to the Hume-Rothery rules, for substitutional solid solutions, the solute and solvent should have similar electronegativity. If there is a large difference of the electronegativity between two elements, the metals tend to form intermetallic compounds instead of solid solutions. For halogens and noble gases, their electronegativities are much higher than Fe at relative low pressures, leading to positive formation enthalpies at 20 and 150 GPa. With increasing pressure, the electronegativity differences between Fe and impurity atoms gradually decreased, and their alloys thus tend to be stable. The lighter halogen atom has a higher electronegativity, leading to a larger difference with Fe, and the behavior of formation enthalpy for the halogens moves down in the periodic table. Furthermore, due to the higher electronegativity of light halogens and noble gas atoms, they are less likely than the heavy atoms to alloy with Fe, which might offer an explanation for the gradual decrease in formation enthalpies of halogens and noble gases when moving down in the periodic table.”

23) *Of what exactly?*

Response: For clarity, we've moved the list of these elements to a suitable position in this sentence.

“Finally, we chose 13 elements as representative systems (P, S, As, Ge, Se, Te, Si, Sb, Ga, Br, Sn, Cl and Bi) to investigate the elastic constants of the expected structures (Table S1).”

24) *Not all these elements are volatiles. Again unclear the selection criteria for the studied elements*

Response: We added the selection criteria of these 13 elements in the revised manuscript.

“These elements are selected from Figure S3, which includes a wide range elements (as shown in Table S4). The Fe alloys with these elements have the lowest formation enthalpies at 300 GPa, indicating that they could be potentially stored in the Earth's inner core, based on our results.”

25) *Which values exactly were used here?*

Response: We added the values of density deficit (~10% at outer core and 3.6% at inner core boundary) and cited the following references in revision.

1. Birch, F. Density and composition of mantle and core. *J. Geophys. Res.* 69, 4377 (1964).
2. Birch, F. Elasticity and constitution of the Earth's interior. *J. Geophys. Res.* 57, 227 (1952).
3. Fei, Y., et al., Thermal equation of state of hcp-iron: Constraint on the density deficit of Earth's solid inner core. *Geophys. Res. Lett.* 43, 6837 (2016).

26) *What is the PREM value? It would be useful to have it to check Fig. 4b.*

Response: The PREM value of V_s in the inner core is about 3.7 km/s and that of pure *hcp*-Fe is about 4.5 km/s (360 GPa, 6600K). In the revised manuscript, we have added the following sentence:

“The S-wave velocities in the inner core (~3.7 km/s) observed from seismology are lower than pure *hcp*-Fe obtained from experimental and theoretical results (~4.5 km/s).”

27) *The y-axis in Figures A and B should be the same for comparative purposes. Also, please add the abbreviations to your scales (also in supplementary figures). For example, the Y-axis label states "Shear wave velocity (V_s , km/s) as in the text; you usually refer to it in abbreviations.*

Response: We have made modifications to Figure 4.

28) *Again several of the elements used in the study are not really considered lithophile, e.g., Se, Te and S.*

Response: In the revised manuscript, we have replaced the “lithophile” from the previous version of manuscript with “lithophile or chalcophile”.

Reviewer #2 (Remarks to the Author):

Response to the reviewer #2

Comments from the Reviewer #2: Overall thoughts:

This was a large, comprehensive study examining the thermodynamic stability of substitutional iron alloys. The work is novel in that it is the first first-principles study to examine such a wide array of elements being incorporated into the hcp-Fe structure with this substitution mechanism. The amount of data is commendable, the methods are sound, and the work will have particular interest for the inner core community. However, the paper is missing a key discussion of the role of liquid iron in their story, particularly during core-mantle differentiation when the entire planet was molten. In general, the paper could use more detailed discussion of the implications of their results for the evolution of the Earth as a whole. More detailed comments are attached below.

Summary: Certain volatile and non-siderophile elements, such as Se and Br, are over-depleted in the silicate Earth compared to the established volatility trend. In this work, the authors perform first-principles simulations examining whether these elements can be incorporated into the Earth's core as substitutional alloys. They find that these alloys are thermodynamically stable, highlighting a potential reservoir for volatile elements within the Earth's deep interior.

Response: We would like to thank this reviewer for the positive evaluation and important suggestions, which have helped us improve the manuscript. We have addressed all comments raised and revised our manuscript accordingly.

Comments from the Reviewer #2: Comments:

1) *The introduction would be improved by including a quick overview of previous work in the field (both experimental and computational) on iron alloys in the inner core.*

Response: We thank this reviewer for this suggestion. We have added an overview of previous work about Fe alloys in the Earth's core in the **Introduction** section and cited some relevant references in the revised manuscript as well.

“It is generally supposed that the low shear velocities (V_S) observed in seismic data from the core imply an inner core density lower than that of pure iron. This suggests the existence of ‘light’ elements as the major alloying candidates within it, such as H, C, O, S and Si..”

[1] Zhu Mao, Jung-Fu Lin, Jin Liu, Ahmet Alatas, Lili Gao, Jiyong Zhao and Ho-Kuang Mao, Sound velocities of Fe and Fe-Si alloy in the Earth's core, Proc. Natl. Acad. Sci. 109, 10239 (2012).

[2] Jung-Fu Lin, Dion L. Heinz, Andrew J. Campbell, James M. Devine and Guoyin Shen, Iron-silicon alloy in Earth's core?, Science 295, 313 (2002).

[3] Jung-Fu Lin, Viktor V. Struzhkin, Wolfgang Sturhahn, Eugene Huang, Jiyong Zhao, Michael Y. Hu, Ercan E. Alp, Ho-kwang Mao, Nabil Boctor and Russell J. Hemley, Sound velocities of iron-nickel and iron-silicon alloys at high pressures, Geophys. Res. Lett. 30, 21 (2003).

- [4] Daniele Antonangeli, Julien Siebert, James Badro, Daniel L. Farber, Guillaume Fiquet, Guillaume Morard and Frederick J. Ryerson, Composition of the Earth's inner core from high-pressure sound velocity measurements in Fe–Ni–Si alloys, *Earth Planet. Sci. Lett.* 295, 292 (2010).
- [5] Nico de Koker, Gerd Steinle-Neumann and Vojtěch Vlček, Electrical resistivity and thermal conductivity of liquid Fe alloys at high P and T, and heat flux in Earth's core, *Proc. Natl. Acad. Sci.* 109, 4070 (2012).
- [6] D. Alfe, M. J. Gillan, and G. D. Price, Composition and temperature of the Earth's core constrained by combining ab initio calculations and seismic data, *Earth Planet. Sci. Lett.* 195, 91 (2002).
- [7] H. Terasaki, S. Kamada, T. Sakai, E. Ohtani, N. Hirao, and Y. Ohishi, Liquidus and solidus temperatures of a Fe–O–S alloy up to the pressures of the outer core: Implication for the thermal structure of the Earth's core, *Earth Planet. Sci. Lett.* 304, 559 (2011).
- [8] G. Morard, D. Andrault, D. Antonangeli, Y. Nakajima, A.L. Auzende, E. Boulard, S. Cervera, A. Clark, O.T. Lord, J. Siebert, V. Svitlyk, G. Garbarino and M. Mezouar, Fe–FeO and Fe–Fe₃C melting relations at Earth's core–mantle boundary conditions: Implications for a volatile-rich or oxygen-rich core, *Earth Planet. Sci. Lett.* 473, 94 (2017).
- [9] Bin Chen, Zeyu Li, Dongzhou Zhang, Jiachao Liu, Michael Y. Hu, Jiyong Zhao, Wenli Bi, E. Ercan Alp, Yuming Xiao, Paul Chow and Jie Li, Hidden carbon in Earth's inner core revealed by shear softening in dense Fe₇C₃, *Proc. Natl. Acad. Sci.* 111, 17755 (2014).
- [10] C. Prescher, L. Dubrovinsky, E. Bykova, I. Kupenko, K. Glazyrin, A. Kantor, C. McCammon, M. Mookherjee, Y. Nakajima, N. Miyajima, R. Sinmyo, V. Cerantola, N. Dubrovinskaia, V. Prakapenka, R. Rüffer, A. Chumakov and M. Hanfland, High Poisson's ratio of Earth's inner core explained by carbon alloying, *Nat. Geosci.* 8, 220 (2015).
- [11] Izumi Mashino, Francesca Miozzi, Kei Hirose, Guillaume Morard and Ryosuke Sinmyo, Melting experiments on the Fe–C binary system up to 255 GPa: Constraints on the carbon content in the Earth's core, *Earth Planet. Sci. Lett.* 515, 135 (2019).
- [12] Lili Gao, Bin Chen, Michael Lerche, Esen E. Alp, Wolfgang Sturhahn, Jiyong Zhao, Hasan Yavas and Jie Li, Sound velocities of compressed Fe₃C from simultaneous synchrotron X-ray diffraction and nuclear resonant scattering measurements, *J. Synchrotron Rad.* 16, 714 (2009).
- [13] Yoichi Nakajima, Eiichi Takahashi, Nagayoshi Sata, Yu Nishihara, Kei Hirose, Ken-ichi Funakoshi and Yasuo Ohishi, Thermoelastic property and high-pressure stability of Fe₇C₃: Implication for iron-carbide in the Earth's core, *Am. Mineral.* 96, 1158 (2011).
- [14] Yuki Shibazaki, Eiji Ohtani, Hiroshi Fukui, Takeshi Sakai, Seiji Kamada, Daisuke Ishikawa, Satoshi Tsutsui, Alfred Q.R. Baron, Naoya Nishitani, Naohisa Hirao and Kenichi Takemura, Sound velocity measurements in dhcp-FeH up to 70 GPa with inelastic X-ray scattering: Implications for the composition of the Earth's core, *Earth Planet. Sci. Lett.* 313, 79 (2012).
- [15] Razvan Caracas, The influence of hydrogen on the seismic properties of solid iron, *Geophys. Res. Lett.* 42, 3780 (2015).
- [16] James Badro, Guillaume François Guyot, Eugene Gregoryanz, Florent Occelli, Daniele Antonangeli and Matteo d'Astuto, Effect of light elements on the sound velocities in solid iron: Implications for the composition of Earth's core, *Earth Planet. Sci. Lett.* 254, 233 (2007).
- [17] Qingyang Hu, Duck Young Kim, Wenge Yang, Liuxiang Yang, Yue Meng, Li Zhang and Ho-Kwang Mao, FeO₂ and FeOOH under deep lower-mantle conditions and Earth's oxygen-

hydrogen cycles, Nature 534, 241 (2016).

2) *In the introduction, I agree that the positive correlation of the formation enthalpies of the Fe-alloys with the abundances of the non-siderophile and volatile elements indicates that these elements could be sequestered in the solid inner core today. However, what seems to be missing is the role that **liquid iron** plays in this story. Since inner core crystallization occurred after differentiation, the abundance of non-siderophile and volatile elements in the bulk silicate Earth would be primarily set by the partitioning of these elements between silicate and liquid metal. Incorporation of these elements as substitutional alloys in the inner core would occur later, affecting their abundances in the liquid outer core. Drawing a direct link between these two processes is confusing.*

Response: We thank this reviewer for the good suggestion. The study of the reaction between liquid iron and an element is quite difficult, and we more focused on the incorporation of a element as substitutional alloy in the inner iron core in the current work. In response, we have added the discussion about the liquid Fe in the revised **Introduction**.

“It is essential to understand the reactions between liquid iron and non-siderophile or volatile elements under high pressures, since the entire primordial Earth was expected to be molten during core-mantle differentiation. However, computational simulations of such conditions are extremely challenging, and more attention has, therefore, been paid to the study of reactions between solid iron and elements depleted in the silicate earth, e.g. Xe, which is relevant to the understanding of the inner core crystallization after core-mantle differentiation.”

3) *Were these calculations spin-polarized? Please indicate in the methods section.*

Response: We thank this reviewer for the good suggestion. In response, we have performed additional spin-polarized simulations for Fe-Si system (Fe₅₃Si) as a representative example. The results suggest that magnetic moment of iron is 0 at high pressures. Moreover, the enthalpy without spin-polarized simulation is slightly lower than that with spin-polarized simulation as shown in the table below, which indicate the simulations including spin-polarized consideration do not alter our conclusion. We have added the details in the revised manuscript accordingly.

	20 GPa (eV/atom)	150 GPa (eV/atom)	300 GPa (eV/atom)
non-spin-polarized	-6.9602	-0.0756	6.6445
spin-polarized	-6.9587	-0.0755	6.6445

“We have performed additional simulations to explore any influence of the spin-polarized effect for Fe. Our results show that there is no spin polarization for Fe over a pressure range of 20-300 GPa (Table S3)”

4) *For easier comparison, it would be helpful to add a dotted line at 0 in Figure S3 (like you did in Figure S4), and to plot the points from Figure S3 on Figure S4 (faded, or in a different color).*

Response: We thank this reviewer for the suggestion. We have updated the figures as the reviewer suggested.

5) *Please cite which geotherm you are using in line 121.*

Response: We thank this reviewer for the comments. We have cited the reference as follows in the revision.

Anzellini S, Dewaele A and Mezouar M et al. Melting of iron at Earth's inner core boundary based on fast X-ray diffraction. Science 2013; 340: 464–6.

6) *In Figure S5, what do you mean by “the fitting of the data?” What exactly was fit? The meaning of the dashed red line is unclear.*

Response: The trend of the dots shown in Figure S5 is not as clear as that in Figure 1, so we performed a linear fit on them. We mainly focus on the slope of this fitted line: a positive slope indicates a positive correlation between the depletion of elements and the stability of the alloy, while a negative slope suggests the opposite. The slope of the fitting line is 6.5×10^{-4} , 0.017 and 0.028 for 20, 150 and 300 GPa, respectively. In response, we have added the explanation of the dashed red line in the revised Supplementary Information.

7) *In general, the figures are quite small and their resolution could be improved.*

Response: We thank this reviewer for the comments. We have improved the resolution of the figures and uploaded them in the system.

8) *While the trend in figure 1 is evident, especially at higher pressures, the spread between different elements is quite high, indicating other factors are coming into play. Adding possible explanations for outliers in the expected trend (such as Li, Mg, B) would be helpful here in your discussion of the figure.*

Response: We thank this reviewer for the good suggestion. We attribute the differences among these elements to the fact that their siderophilicities vary differently with pressure. We have added the following discussion in the revised **Results** section.

“Some elements, such as S, Se and Te, are seen to become more siderophile with increasing pressure. In Figure 1, the formation enthalpies of alloys Fe_{127}S , Fe_{127}Se and Fe_{127}Te decrease significantly (~ 0.05 eV/atom) as the pressure increases from 20 GPa to 300 GPa. Our simulations are thus in agreement with the previous results, which is also a validation of computational scheme of the current study. Furthermore, we also noticed that, with increasing pressure, the formation enthalpies of several alloys, such as Fe_{127}Li and Fe_{127}B , decrease slightly (~ 0.02 eV/atom), indicating that the siderophilicities of these impurity elements will not have changed significantly during core segregation.”

9) *For comparison purposes in Figure 2, make the y axis scale the same for all plots.*

Response: We thank this reviewer for the good suggestion. In response, we have updated the Fig. 2 with the same y axes.

10) *Do you have an explanation for the different formation enthalpy behavior of the halogens as you move down the periodic table? They are not included in the discussion of Figure 2.*

Response: We thank this reviewer for the good suggestion. According to the Hume-Rothery rules, for substitutional solid solutions, the solute and solvent should have similar electronegativity. If the electronegativity difference is large, the metals tend to form intermetallic compounds instead of solid solutions. For halogens, their electronegativities are

much higher than Fe, resulted in the positive formation enthalpies at 20 and 150 GPa (Xiao Dong, et al., Electronegativity and chemical hardness of elements under pressure. Proc. Natl. Acad. Sci. U.S.A. 119, 10 (2022)). The lighter halogen atom has a higher electronegativity, leading to a greater difference with Fe, so the formation enthalpy behavior of the halogens moves down the periodic table. In response, we have added the explanation in the revised manuscript and expanded the discussion part.

“As observed in Figure 2, the formation enthalpy behaviors of the halogens (F, Cl and Br) and noble gases (Ne, Ar, Kr and Xe) decrease with increasing mass, which is related to their high electronegativities under both ambient and high pressure. According to the Hume-Rothery rules, for substitutional solid solutions, the solute and solvent should have similar electronegativity. If there is a large difference of the electronegativity between two elements, the metals tend to form intermetallic compounds instead of solid solutions. For halogens and noble gases, their electronegativities are much higher than Fe at relative low pressures, leading to positive formation enthalpies at 20 and 150 GPa. With increasing pressure, the electronegativity differences between Fe and impurity atoms gradually decreased, and their alloys thus tend to be stable. The lighter halogen atom has a higher electronegativity, leading to a larger difference with Fe, and the behavior of formation enthalpy for the halogens moves down in the periodic table. Furthermore, due to the higher electronegativity of light halogens and noble gas atoms, they are less likely than the heavy atoms to alloy with Fe, which might offer an explanation for the gradual decrease in formation enthalpies of halogens and noble gases when moving down in the periodic table.”

11) The discussion would benefit from an additional paragraph at the end, explaining in more detail the implications of your results for the thermochemical evolution of the Earth. For example, discussion of the expected composition of the inner core, outer core, and mantle from your results and the effect this has on physical and transport properties.

Response: We thank this reviewer for his/her constructive suggestions which highly improves the quality of our manuscript. We have added the comparative results with previous studies and extrapolated the static results to high temperatures.

“The densities, wave velocities and compressibility of iron-rich alloys under high P–T conditions, compared with observed seismic models, such as PREM, serves as an important constraint on the Earth’s core composition. Previous studies have primarily focused on ‘light’ elements, as candidate impurities of the Fe alloys. Our results suggest that certain non-siderophile and volatile elements could be potentially become captured into the Earth's core during metal–silicate segregation, to enrich the chemical composition of the core. Compared with the relationship between static wave velocities and high temperature V_s in the systems Fe–C, Fe–O and Fe–Si reported previously, we extrapolated the trend of high temperature V_s for the elements shown in Figure 4. The S -wave velocities in the inner core observed from seismology (~ 3.7 km/s) are lower than pure hcp -Fe obtained from experimental and theoretical results (~ 4.5 km/s). To extrapolate our static findings to elevated temperatures, we compared the calculated V_s of pure hcp -Fe with that of Fe alloys in previous works at high temperatures and 0 K in Figure S9. The data from previous studies (dashed lines) suggest that ‘light’ atoms are likely to soften the static V_s of hcp -Fe, and this effect remains similar at elevated temperatures. This trend indicates that the doping of ‘heavy’ elements we have considered at 0 K (red open squares)

in Figure 4a may have the potential to decrease V_s of *hcp*-Fe at high temperatures. As doping concentrations increase, the static V_s of the alloys linearly decreases (Figure 4b), suggesting that high temperature V_s of Fe alloys may be decreased due to doping of these heavier impurity atoms. Our current findings thus suggest the estimation of the concentration of ‘light’ elements within the Earth's mantle as well as the Earth’s core remain an open challenge.”

REVIEWERS' COMMENTS

Reviewer #1 (Remarks to the Author):

After carefully examining the revised manuscript by Tian et al., I am pleased to note that the authors have thoroughly addressed all of my initial comments and concerns. While I have identified a few minor issues and suggested some areas for improvement (see attached file), such as clarifying certain sections and ensuring accurate table references, these are relatively minor adjustments. Specifically, the sentences from lines 87 to 95 could benefit from rephrasing, as they currently seem somewhat confusing, possibly due to the recent inclusion of new information. This has resulted in a back-and-forth discussion between elements that can be explained by the presence of Fe and those that cannot.

Once these minor comments have been addressed, I recommend accepting the manuscript for publication, pending consensus from all reviewers.

Reviewer #2 (Remarks to the Author):

The authors have done a great job addressing my concerns in revising the manuscript. The new additions add clarity, provide additional helpful background for the reader, and flesh out the implications of the work.

Reviewer #1 (Remarks to the Author):

Response to the reviewer #1

Comments from the Reviewer #1: *After carefully examining the revised manuscript by Tian et al., I am pleased to note that the authors have thoroughly addressed all of my initial comments and concerns. While I have identified a few minor issues and suggested some areas for improvement (see attached file), such as clarifying certain sections and ensuring accurate table references, these are relatively minor adjustments. Specifically, the sentences from lines 87 to 95 could benefit from rephrasing, as they currently seem somewhat confusing, possibly due to the recent inclusion of new information. This has resulted in a back-and-forth discussion between elements that can be explained by the presence of Fe and those that cannot.*

Once these minor comments have been addressed, I recommend accepting the manuscript for publication, pending consensus from all reviewers.

Response: We appreciate this reviewer for his/her careful reading and positive comments. We made minor adjustments to the manuscript based on the reviewers' suggestions, including revisions to the references, polishing of certain sentences, and formatting of Supplementary Information. We wish our manuscript is acceptable for publication at the current form.

Reviewer #2 (Remarks to the Author):

Response to the reviewer #2

Comments from the Reviewer #2: *The authors have done a great job addressing my concerns in revising the manuscript. The new additions add clarity, provide additional helpful background for the reader, and flesh out the implications of the work.*

Response: We would like to appreciate this reviewer for his/her positive evaluation.